# The Development of the *Chimaeroid* Pelvic Skeleton and the Evolution of Chondrichthyan Pelvic Fins

**DOI:** 10.3390/jdb10040053

**Published:** 2022-12-12

**Authors:** Jacob B. Pears, Carley Tillett, Rui Tahara, Hans C. E. Larsson, Kate Trinajstic, Catherine A. Boisvert

**Affiliations:** 1School of Molecular and Life Sciences, Curtin University, Perth, WA 6102, Australia; 2Curtin Health Innovation Research Institute, Curtin University, Perth, WA 6102, Australia; 3Hub for Immersive Visualisation and eResearch, Curtin University, Perth, WA 6102, Australia; 4Redpath Museum, McGill University, Montréal, QC H3A 0C4, Canada; 5Western Australian Museum, Kew St, Welshpool, WA 6000, Australia

**Keywords:** chimaeroid, pelvic girdle, skeletal development, cladoselache, chondrichthyes

## Abstract

Pelvic girdles, fins and claspers are evolutionary novelties first recorded in jawed vertebrates. Over the course of the evolution of chondrichthyans (cartilaginous fish) two trends in the morphology of the pelvic skeleton have been suggested to have occurred. These evolutionary shifts involved both an enlargement of the metapterygium (basipterygium) and a transition of fin radial articulation from the pelvic girdle to the metapterygium. To determine how these changes in morphology have occurred it is essential to understand the development of extant taxa as this can indicate potential developmental mechanisms that may have been responsible for these changes. The study of the morphology of the appendicular skeleton across development in chondrichthyans is almost entirely restricted to the historical literature with little contemporary research. Here, we have examined the morphology and development of the pelvic skeleton of a holocephalan chondrichthyan, the elephant shark (*Callorhinchus milii*), through a combination of dissections, histology, and nanoCT imaging and redescribed the pelvic skeleton of *Cladoselache kepleri* (NHMUK PV P 9269), a stem holocephalan. To put our findings in their evolutionary context we compare them with the fossil record of chondrichthyans and the literature on pelvic development in elasmobranchs from the late 19th century. Our findings demonstrate that the pelvic skeleton of *C. milii* initially forms as a single mesenchymal condensation, consisting of the pelvic girdle and a series of fin rays, which fuse to form the basipterygium. The girdle and fin skeleton subsequently segment into distinct components whilst chondrifying. This confirms descriptions of the early pelvic development in *Scyliorhinid* sharks from the historical literature and suggests that chimaeras and elasmobranchs share common developmental patterns in their pelvic anatomy. Alterations in the location and degree of radial fusion during early development may be the mechanism responsible for changes in pelvic fin morphology over the course of the evolution of both elasmobranchs and holocephalans, which appears to be an example of parallel evolution.

## 1. Introduction

One of the morphological novelties to arise in the jawed vertebrates (gnathostomes), is the pelvic skeleton, comprising the pelvic girdle, paired pelvic fins [1,2,3], and intromittent organs (claspers) [1,4]. The gnathostomes (jawed vertebrates) comprise two extant clades, the Chondrichthyes (Elasmobranchii and Holocephali) and Osteichthyes and, the extinct placoderms, considered either an array of paraphyletic stem gnathsotomes [5] or as the sister clade to all other gnathosotmes gnathostomes [6,7] and the extinct Acanthodians most recently resolved as stem chondrichthyans [7,8]. An evolutionary transition in the morphology of the chondrichthyan pelvic skeleton has been suggested to have occurred over the course of the evolution of the Chondrichthyes by previous authors [4]. This transition is supposed to have involved an enlargement of the basipterygium and the migration of the pelvic fin radial articulation from the girdle to the basipterygium [4]. However, since this hypothesis was postulated, there has been significant revision in the phylogeny of chondrichthyans [8,9,10,11] and the developmental mechanisms underpinning these changes are unknown.

One means of investigating the mechanisms of morphological change observed in the fossil record is to examine the development of the anatomy of extant taxa [12,13]. The Chondrichthyes have long been used as models to examine the evolution and development of the appendicular anatomy [14,15,16], often under the false notion that they are primitive, and that their morphology represented the “ancestral” body plan [14,15,17,18]. Current understandings of the anatomy and fossil record of Chondrichthyans indicate that these taxa possess a mixture of derived and plesiomorphic morphologies which can be examined to investigate the evolution of different traits across the gnathostome clade when compared with extant and fossil taxa [4,19,20]. Studies of the morphology of the chondrichthyan appendicular skeleton development are largely restricted to the late 19th [14,17,21,22,23] and early 20th centuries [24,25,26] with contemporary studies almost exclusively focusing on molecular aspects of early development rather than morphology [27,28,29]. As highlighted by previous authors [30,31], the scarcity of current studies on the morphology of the development of the endoskeleton of chondrichthyans represents a notable knowledge gap, and this lack of fundamental data to infer plesiomorphic characters through comparative anatomy and development may inhibit the study of the evolution of the skeletons of jawed vertebrates.

The Holocephali are one of the two extant clades of the Chondrichthyes [8,11]. Recently the morphology of the pectoral fins of *Cladoselache*, a stem holocephalan, has been revised [32]; however, despite the availability of more complete pelvic material, a re-description has not taken into account features acknowledged to be misinterpreted in historical descriptions [33,34,35]. Chimaeroids are the only extant holocephalan chondrichthyans [8,36] and may serve as good models to investigate the evolution of the pelvic skeleton as they retain unfused pelvic girdles [37,38], which is considered the plesiomorphic condition of chondrichthyans [39,40]. There is only one morphological study of the chimaeroid appendicular skeleton development, that of the elephant shark (*Callorhinchus milii*), using wholemount clearing and alcian blue staining [41]. That study focused on the serial homology of the paired fins, not including the development of the pelvic girdle [41]. 

Here, we present descriptions of the adult morphology and development of the pelvic and reproductive skeleton of *C. milii* using a combination of histology, traditional and digital dissection (segmentation of micro and nanoCT scans) of adult and embryonic specimens. The use of nanoCT scanning allows for the complete visualisation of the pelvic skeleton development in three dimensions, during early development, which is not otherwise visible using clearing and staining techniques. In order to identify the cellular identities and growth processes of the pelvic fin skeleton, the 3D dataset is complemented by haematoxylin, eosin and alcian stained histological sections. We also redescribe the pelvic skeleton of *Cladoselache* and map pelvic morphology onto a current chondrichthyan phylogeny [8,9] using existing descriptions of pelvic skeleton morphology. 

## 2. Materials and Methods

### 2.1. Materials

#### 2.1.1. Fossil Material

*Cladoselache kepleri* (NHMUK PV P 9269), collected from the Cleveland Shale Member of the Ohio Shale Formation (Late Devonian) in Ohio, USA and housed in the fossil fish collections of the Natural History Museum (London, UK). It consists of a whole body specimen, preserved in a supine position (Figure 1A). This specimen has been examined by previous authors [34,42,43] and was noted for having relatively well preserved pelvic fins compared to most *Cladoselache* fossil [42,44] (Figure 1A).

#### 2.1.2. Adult & Embryonic Materials

The embryos forming the growth series examined here were sourced from adult female *Callorhinchus milii* caught by rod and reel from Western Port Bay, Victoria Australia (Licence Numbers: DPR RP1000, RP1003 and RP1112). The transport, husbandry and release of caught females followed established practice [45]. Eggs laid in captivity were incubated in a closed system aquarium [45] and embryos were euthanized with Tricaine at different stages of development. Embryonic specimens were fixed in 4% paraformaldehyde (PFA) in phosphate-buffered solution (PBS) and dehydrated in either ethanol or methanol. These procedures were conducted in accordance with the authorisation and directions of the Monash University Animal Ethics Committee (Permit: MAS/ARMI/2010/01).

Some of the adult male and female specimens, also caught by rod and reel Licence Numbers: DPR RP1000, RP1003 and RP1112) and treated with the same practices [45], died in captivity and were used for dissections. These actions were also conducted with the authorisation and direction of the Monash University Animal Ethics Committee (Permit: MAS/ARMI/2010/01).

### 2.2. Methods

#### 2.2.1. Skeletonisation and Gross Anatomy

The pelvic region of adult male and female *C. milli* was skeletonised using hot water (50–75 °C) and forceps to study the morphology of the adult pelvic skeleton in both sexes. 

#### 2.2.2. Histology

To examine the cellular aspects of the growth and development of the pelvic skeleton, the pelvic area from a growth series of embryos (stages 30, 31, 32 & 34 [46]) was fixed using 4% PFA in PBS. Specimens were embedded in paraffin wax using a Leica TP1020 tissue processor with the following setup: two hours each in 70% ethanol in PBS, 80% ethanol in PBS, 96% ethanol in PBS, 100% ethanol in PBS twice, Xylene three times, and twice in paraffin. Tissue blocks were sectioned using a Leica RM2235 rotary microtome (section thickness: 10 µm) and floated and dried in a Wuhan Junjie JK-6 flotation work station (Float and Dry 42 °C). Paraffin sections were dewaxed and cleared with xylene and stained with haematoxylin, eosin and alcian blue (0.2% alcian blue in 40% ethanol; pH 3.6). Slides were imaged using a Zeiss Axio Scan Z.1 slide scanner at 10× magnification with the brightfield 10× scanning profile (BF_10x_Default_ZMB). 

#### 2.2.3. NanoCT Imaging

To examine the development of the skeletal elements across ontogeny in situ, a growth series (stages 30, 32 and 34 [46]) of elephant shark (*Callorhinchus milii*) embryos were stained and scanned using nanoCT imaging (Zeiss Xradia 520 Versa, 60 kV). This method allowed for non-destructive examination of the embryonic material. The embryos were stained using 1% phosphotungsitc acid [PTA] (1% PTA dissolved in 70% EtOH). The specimens were quickly washed with 70% ethanol to remove excessive stain on the surface, and then scanned in an optimal medium (0.5% agarose, or 70% EtOH—larger sample) (For the staining period and imaging parameters for each stand see Table 1).

#### 2.2.4. MicroCT Imaging

To examine the morphology of an adult male, half of a bisected trunk of an adult male *C. milli* was stained with a 1% solution of Iodine in ethanol for 3 months and scanned in a Skyscan 1176 micro-CT scanner. Specimens were removed from the iodine solution and wrapped in ethanol soaked cotton towels prior to imaging (For imaging parameters see Table 1).

#### 2.2.5. 3D Modelling

NanoCT and microCT data were reconstructed using the software associated with the respective instruments. The skeletal elements were segmented and visualised using *Dragonfly* version 2022.1 (Object Research Systems).

#### 2.2.6. Terminology

The terminology used for the descriptions of the pelvic girdle, fin and clasper cartilages of *C. milii* follows that of Didier [38] and Riley and colleagues [41]. The terminology for the components of the pre-pelvic tenaculum is taken from the description of Leigh-Sharpe [47].

#### 2.2.7. Language of Descriptions

Structures and processes are described in the CT and histology data sets as a continuum, though distinct dead individuals from different stages of development are being used. This choice was taken as this form of language makes it easier for the reader to understand the descriptions and we justify the description of growth, mitosis and other processes through the morphological evidence presented and with the understanding that these are static in the dead specimens.

## 3. Results

### 3.1. The Pelvic Skeleton of Cladoselache

NHMUK PV P 9269 (Figure 1A,B) has two mesially situated pelvic girdles, the left girdle being better preserved. The girdles have an elongated triangular shape with the longest process of each being situated anteriorly. Whilst these processes are oriented towards each other proximally, they lack a symphysis. Posterior to the left girdle is a subtriangular metapterygial element which tapers off posteriorly. A less well preserved metapterygium is preserved posterior to the right girdle. Perpendicular to the distal sides of each girdle and metapterygium is a series of fin radials, which exhibit a centripetal variation in length and breadth, being longer and thicker in the centre of the fins and shorter towards the anterior and posterior extents of the fins. Different numbers of radials are preserved in each fin, 15 in the right and 12 in the left. The eight most anterior radials of each fin are in close proximity with the pelvic girdle, with the remaining radials articulating directly on each metapterygial element or being in close proximity to them. Due to some poor preservation and damage to the specimen, the full extent of the pelvic girdles and their relationship with the fin radials and metapterygia is not certain (Figure 1A). Thin skeletal elements of uncertain identity are located between some of the posterior fin radials, potentially representing ceratotrichia [33] or intercalary radials [32,48].

### 3.2. Adult Morphology of Callorhinchus milii

A microCT scan of an adult male (Figure 2) and the skeletonization of adults of both sexes (Figure 3) show that in addition to the central vacuity the pelvic girdle is perforated by five other foramina: two immediately posterior to the central vacuity and three at near the base of iliac process (Figure 2 and Figure 3). The anterior clasper cartilage is composed of three distinct components rather than a single entity. It consists of a primary cartilage, articulating with the basipterygium and posterior clasper cartilage, and two secondary cartilages. The larger of these secondary components is a cartilaginous arch that articulates on the dorsal surface of distal side of the primary component. We propose that this element be called the distal secondary component. The other is a small triangular cartilage embedded on the proximal side of the primary cartilage, which we propose be called the proximal secondary component.

### 3.3. Developmental Data

#### 3.3.1. Stage 30

##### NanoCT

The pelvic skeleton consists of a single condensation of tissue in which rudiments of the pelvic girdle, basipterygium and fin radials can be identified (Figure 4A,B). The pelvic girdle resembles a reduced form of the adult skeleton with a shorter, thinner iliac process and a shorter anterior process that does not extend as far anteriorly over the intestines and lacks a central vacuity. The fin skeleton consists of a series of radials, fanning across and extending distally within the fin bud, exhibiting an antero-posterior gradient of density. Tissue is concentrated between and along these radials forming the rudimentary basipterygium. The basipterygium has already taken on a disk like shape reminiscent of the adult. The two most anterior radials are in direct contact with the pelvic girdle, forming the articulation between the basipterygium and the girdle. The portions of the radials distal to the basipterygium are the rudiments of the middle fin radials. Whilst the radials do not extend very far within the fin, very diffuse tissue at their distal margins (Figure 4A,B) resembles the distal fin radials of the adult (Figure 2 and Figure 3). A posterior extension of tissue can be observed projecting from the pelvic girdle (Figure 4B). The identity of this structure is uncertain as it is not present at later stages and does not correspond with any known skeletal element.

##### Histology

The skeleton is difficult to detect histologically, however histogenesis of the pelvic girdle and basipterygium can be observed. Clusters of mesenchymal cells have formed rudiments of the pelvic girdle and basipterygium in the form of a single mesenchymal condensation (Figure 4C–H). The condensation of the anterior process is located beneath and parallel with the intestines (Figure 4C–F, PG). Slightly posteriorly, this portion of the condensation extends dorsally forming the iliac process, which only just sits above the hypaxial musculature. The basipterygium extends within the fin (Figure 4F–H, B), but there is no evidence of the histogenesis of the fin radials, pelvic clasper or tenaculum cartilages.

#### 3.3.2. Stage 31

##### Histology

The pelvic girdle and fin skeleton still consist of a single condensation but most of the individual skeletal elements are now clearly identifiable histologically (Figure 5). Each element displays a varied level of development and cellular morphology. Condensations of tightly packed rounded cells have partly formed much of the pelvic girdle, particularly the anterior process beneath and parallel with the intestines, and the iliac process over the hypaxial musculature (Figure 5A–E, PG). The anterior process and the base of the iliac process are relatively thick and composed of tightly grouped rounded cells proximally and smaller flat cells distally (Figure 5C,D, PG). The dorsal portion of the iliac process is more diffuse being composed of loosely distributed rounded cells and many small flat cells (Figure 5A,B, PG). Similarly, the most proximal and anterior parts of the anterior process are more diffuse and formed by flatter cells (Figure 5C–F, PG). The girdle already possesses the five vein foramina (Figure 5A–C, F) and the central vacuity has begun to take shape (Figure 5A, CV). Rudiments of the pre-pelvic tenacula have begun to form, but specific components are not identifiable (Figure 5A, TR).

The fin skeleton is more defined relative to stage 30 and most of its elements, the basipterygium and fin radials, can be identified, being composed of closely clustered rounded cells, which are flatter centrifugally (Figure 5C–H, B, FR). The basipterygium has formed much of its span within the fin bud but is diffuse or absent posteriorly, particularly near the basipterygium foramina, which has partially formed (Figure 5F,F). The basipterygium condensation divides posteriorly, where it still consists of some fin radials (Figure 5G,H).

A rudimentary primary anterior clasper cartilage has begun to form. This takes the form of a cluster of slightly rounded cells and the surrounding mesenchymal tissue extending from the basipterygium in a proximal direction (Figure 5F–H, AC). These clusters resemble part of the shape of this cartilage at later stages (Figure 6) but are still diffuse, being composed of a small core of clustered cells with some alcian blue staining surrounded by mesenchymal tissue that blends with the undifferentiated connective tissue.

#### 3.3.3. Stage 32

##### NanoCT

The iliac and anterior processes of the pelvic girdle have, respectively, expanded dorsally over the hypaxial muscle and anteriorly over the intestines (Figure 6A,B) but are still proportionally shorter than those of adults (Figure 2 and Figure 3). The anterior process has expanded sufficiently to form the central vacuity. The basipterygium has broadened laterally and thickened relative to stage 30, but still lacks the full span within the fin found in the adult, particularly on its proximal edge near the articulation with clasper cartilage. Most of the medium fin radials have formed and expanded distally within the fin, but still taper off posteriorly and proximally on the basipterygium where they are still forming or absent at this stage. The distal radials are now clearly taking shape at the end of the medium radials and also display an anteroposterior gradient of development, being absent posteriorly. Both clasper cartilages can be identified (Figure 6A,B). The primary anterior cartilage has reached an extent similar to the adult (Figure 2), but does not share the same shape morphology, lacking its secondary components. It is in contact with the posterior cartilage, which has only partially formed (Figure 6B), lacking both the proportions and scroll like morphology of the adult (Figure 2). The pre-pelvic tenaculum has begun to form, but the grappler (sensu [47]) is the only identifiable component among the rudiments.

##### Histology

By stage 32, almost every element of the pelvic and reproductive skeleton can be identified histologically, and most are clearly chondrogenic. The pelvic girdle has broadened at its peripheries relative to stage 31, and has expanded anteriorly beneath and dorsally over the hypaxial muscles (Figure 6C–I, PG). Almost all of the girdle is now chondrogenic, as evidenced by alcian blue staining of the cartilage matrix and the mitotic chondrocytes, pyramidal cells within lacunae, that form most of this structure. Chondrocytes that are situated more toward the interior of the girdle, are larger and more commonly found in closely clustered isogenous groups relative to those situated at the peripheries. These distal chondrocytes are noticeably smaller and flatter and more densely packed, forming a distinct peripheral layer (~20 µm thick). In certain portions of the anterior process of the girdle this peripheral layer has formed an incomplete or immature perichondrium (5–10 µm thick), exhibiting dark pink staining and a gradient of densely packed fibrous cells (Figure 6G,H, PC). The anterior segments of the pelvic girdle forming the rudimentary central vacuity and the pre-pelvic tenacula have more visibly condensed but are not yet chondrogenic, lacking both matrix deposition, indicated by an absence of alcian blue staining, and chondrocytes (Figure 6D,E, PG, CV, TG). These portions of the girdle and tenaculum are composed of cells with a similar morphology and regionalisation as the skeletal elements in stage 31 (Figure 5). The portion of the pelvic girdle near its connection with the basipterygium also displays a different morphology from the rest of the girdle. This region has less matrix deposition, indicated by a lack of alcian blue staining, and the cells occupying it are relatively rounder and/or flatter than the chondrocytes forming the rest of this structure, but following the same organisational variation as the rest of the chondrogenic parts of this structure (Figure 6H, PG).

Most of the components of the fin skeleton, i.e., the basipterygium and some fin radials, have partly formed within the pelvic fin, but not all these elements are chondrogenic. The basipterygium is almost entirely chondrogenic and is only prechondrogenic at its most distal point anteriorly (Figure 5H, B). The chondrocytes forming most of the basipterygium share the same morphology and arrangement as that of the pelvic girdle, with a distinct core and a peripheral layer. The chondrogenic parts of the basipterygium also possess a perichondrium around its articulation with the pelvic girdle and over most of the fin bar, only being absent proximally near its foramina and the anterior clasper cartilage (Figure 6H–K, B, AC). The pre-chondrogenic part of the basipterygium is located at its most anterior point (Figure 6H) where it is distally composed of cells with the same morphology and organisation as the pre-chondrogenic parts of the girdle. At the basipterygium’s most posterior point it displays alcian blue staining but is diffuse and composed of fibrous cells. The most distal fin radials are also prechondrogenic and share the same cell morphology and organisation as the prechondrogenic parts of the basipterygium (Figure 6I–L), though posteriorly some show signs of cartilage matrix, indicated by alcian blue staining (Figure 6L). The chondrogenic fin radials are still dividing from the basipterygium and share the same cell morphology the other chondrogenic parts of the skeleton and also possess perichondria. These radials are connected with the basipterygium and each other by tissue with the same cell morphology as the articular regions of the pelvic girdle. Posteriorly some of the radials, near the clasper cartilages, have started to condense, but they are not chondrogenic, sharing the same morphology as the other prechondrogenic parts of the skeleton (Figure 5L).

The primary anterior clasper cartilage (sensu [38]) is well defined anteriorly near its connection with the basipterygium, sharing the same cell morphology and organisation of the more developed regions of the pelvic girdle and basipterygium (Figure 6I,J, AC). However, the cell morphology of the rest of this structure resembles that of the articular regions of the pelvic girdle, but with more cartilage matrix deposition, as indicated by alcian blue staining, which decreases as this structure extends posteriorly. The posterior clasper cartilage is less well defined, sharing the cell morphology of the pre-chondrogenic parts of the skeleton, becoming more diffuse posteriorly where it has yet to condense (Figure 6K,L, PC). Neither of these structures possess a perichondrium.

#### 3.3.4. Stage 34

##### NanoCT

The pelvic girdle has extended further over the hypaxial musculature forming its most anterior and posterior points (Figure 7A, B), being proportionally similar to the adult (Figure 2). The tenaculum (Figure 7A, B) is in a condition similar to that of stage 32 (Figure 6A,B) with the grappler being the only clearly identifiable component. The basipterygium has expanded in breadth laterally to resemble more closely that of the adult. The middle fin radials have all formed and the majority of the distal radials have also formed but are still less prominent on the posterior and proximal edges of the fin where they are smaller or absent. The clasper cartilages are much thicker and have lengthened posteriorly relative to stage 32. The primary anterior cartilage is denser and clearly defined along its range, more closely resembling that of the adult (Figure 2), though the secondary components are still absent. The posterior cartilage has extended posteriorly and now possesses the scroll like morphology of the adult, but it has still not fully extended posteriorly to the extent of the adult.

##### Histology

All of the pelvic girdle have chondrified except for the portions of the anterior process framing the central vacuity (Figure 7C–E). The cartilage of the pelvic girdle has matured with the chondrocytes forming it being separated by more matrix relative to stage 32 and includes empty lacunae and more isogenous groups (Figure 7C,H, PG). Beyond this decrease in clustering the girdle retains the same internal organisation and cell morphology with a core and peripheral layer and a distinct region of rounded cells at its articulation with the pelvic girdle (Figure 7J). The chondrogenic parts of the girdle all possess a perichondrium composed of fibrous cells, exhibiting a light pink staining, that blend with the adjacent mesenchyme (Figure 7C). The perichondrium is largely uniform across the structure (10–20 µm thick), however the perichondrium over the proximal surface of the iliac process is thicker (25–30 µm) and exhibits darker eosin staining (Figure 7H, PC). The prechondrogenic parts of the girdle framing the central vacuity are still pre-cartilaginous but have condensed more relative to stage 32 (Figure 6). The pre-pelvic tenaculum has also more fully condensed and taken shape, but this structure is still pre-cartilaginous, indicated by a lack of alcian blue staining, and the grappler remains the only identifiable component (Figure 7F).

The entire fin skeleton has condensed and is almost entirely chondrogenic (Figure 7H–L, B, FR). The basipterygium is now completely chondrogenic, sharing the same internal cell morphology as the pelvic girdle, including the shared point of articulation, and is overlayed by a perichondrium (Figure 7H–K, B, PC). Most of the fin radials have thickened relative to stage 32 and exhibit the same cell morphology as the other chondrogenic structures. However, some radials situated in the most posterior portion of the fin, near the posterior clasper cartilage, are only just chondrifying, exhibiting limited alcian blue staining and few chondrocytes (Figure 7K). The radials still possess a connective tissue with the basipterygium and themselves (Figure 7J–L, FR, B) which shares the same morphology as the articulation of the girdle and basipterygium.

The primary anterior clasper cartilage is now chondrogenic, possessing the same cell morphology as the other chondrogenic parts of the skeleton, but with a thicker (30 µm) perichondrium (Figure 7J, AC). The articular surface with the basipterygium shares the same morphology as the articular surface of the girdle, though the cells have a flatter shape. The posterior clasper cartilage has more fully condensed and has partially chondrified anteriorly, as indicated by some alcian blue staining (Figure 7K, PC), but is still largely pre-cartilaginous (Figure 7L, PC).

#### 3.3.5. Stage 36

##### NanoCT

The pelvic and reproductive skeleton now bear the closest resemblance to that of the adult (Figure 8). The pelvic girdle and fin skeleton are essentially the same as that of the adult in miniature. The primary anterior clasper cartilage also more closely resembles the adult with its grooves and features being more well defined compared to stage 34 (Figure 7A,B), but the secondary components of this structure have still not formed. The posterior clasper cartilage has extended further posteriorly and taken on a well-defined scroll like morphology akin to that of the adult. The components of the pre-pelvic tenacula, the grappler, frill, and gland duct (sensu [47]) are now clearly identifiable, but these are much smaller proportionally compared to that of the adult (Figure 2).

## 4. Discussion

### 4.1. The Pelvic Skeleton of Cladoselache

Our re-examination of the pelvic skeleton of *Cladoselche kepleri* (NHMUK PV P 9269) clarifies aspects on the arrangement and anatomy of the pelvic skeleton from description of this specimen by Jaekel [34]. We concur with this description in that the pelvic girdles of *Cladoselache* are distinct entities that do not join mesially [44,48] and that there is a metapterygial element with which some posterior radials articulate, which has also been recognised by Hussakof and Bryant [44]. However, we find that there is no evidence of the propterygium, an additional basipterygium, pterygopodia (claspers), nor proximal radials [34]. We propose that the anterior distal surface of each girdle likely articulated with at least eight of the adjacent fin radials and more posteriorly with a metapterygial element with which approximately seven radials articulated (Figure 1C). This arrangement is similar to the pelvic fin morphology of other members of the Symmoriiformes such as *Akmonistion zangerli* [49], *Falcatus falcatus* [50], and *Damocles serratus* [51] in which the pelvic skeleton comprised a pelvic girdle with which the fin radials primarily articulated and a posterior metapterygium (Figure 9).

### 4.2. The Development of the Pelvic Skeleton in Extant Chondrichthyans

Here, we have documented the development of the pelvic skeleton of *C. milii* (Figure 10). At the earliest point of skeletogenesis examined here in a stage 30 *C. milli* embryo, the pelvic skeleton, the girdle and basipterygium and fin radials, initially consist of a single condensation of mesenchymal cells (Figure 4). This condensation is diffuse and not easily detected with histological staining until stage 31 (Figure 5). However, this condensation can be visualised in detail at stage 30 through nanoCT imaging (Figure 4A,B). Our findings make it clear that by this stage much of the patterning of the pelvic skeleton, particularly the pelvic girdle, has already taken place prior to chondrogenesis. To our knowledge there is only one contemporary study of the morphology of fin skeleton development in cartilaginous fish, examining *C. milii* using clearing and alcian blue staining [41], which examined a stage 29 embryo. At this stage the only visible structures were a series of mesenchymal rods and no significant change occurred by stage 30 [41]. Given the level of skeletal development observed in stage 30 in the present study it is likely that more of the skeleton has formed by stage 29 than just these rods. The morphology of the earlier development of the pelvic skeleton could be determined in future investigations using nanoCT imaging.

Some historical studies of pelvic development of scyliorhinid sharks, whose developmental stages are approximately equivalent with *C. milli* [46,63], agree with our findings on the early development of the pelvic fin and girdle consisting of a single mesenchymal condensation [14,21,22]. For instance, in his study of the development of the skeleton of paired fins of the nursehound (*Scyliorhinus stellaris*) [14], Balfour states that both the pectoral and pelvic fin skeletons are first visible within the “indifferent mesoblast” and are only distinguished histologically from the surrounding cells by being “more concentrated,” and their borders are not “strongly marked,” ([14] p. 665). Both sets of fins are also observed to arise “simultaneously and continuously with the pectoral and pelvic girdles,” ([14] p. 665). Further, in agreement with our findings at stage P (stage 31 [63]) the girdle and basipterygium of *S. stellaris* are still continuous and the fin radials are continuous with the basipterygium. Likewise, in Wiedersheim’s study of three other sharks, the early pelvic skeleton of the small-spotted cat shark (*Scyliorhinus canicula*) and blackmouth catshark (*Pristiurus [Galeus] melastomus*) (15 mm and 19 mm respectively) are also described as consisting of an condensation of cells within the fin mesoderm [21]. In slightly older *S. canicula* embryos the pelvic girdle and “free extremities” are described as a “single coherent mass” ([21] p. 28) and the pelvic fin and girdle are also initially continuous in the Spiny dogfish (*Squalus acanthias*) ([21] p. 30). No observations are made on the condensations of the clasper skeleton.

A point of disagreement between these historical descriptions is the development of the fin skeleton, i.e., the basipterygium and fin radials. Balfour [14] describes the fin skeleton during early development (before stage 31 sensu [63]) in *S. stellaris* as consisting of a “bar [basipterygium]” whose outer surface consists of a “thin plate” extending into the fin that by stage P (stage 31 sensu [63]) has begun to differentiate into fin radials and notes that this occurs before chondrification [14]. In contrast, separate studies of *P. melastomus* by Wiedersheim [22] and *S. canicula* by Dohrn [21] suggest that the basipterygium is formed by union of individual radii or “cartilage rays”. Wiedersheim also states that the pelvic girdle itself is also a product of this fusion [21]. One of these authors, Wiedersheim [21], contends that Balfour’s observations are mistaken as they have been conducted in specimens too old to see this fusion. There are some anatomical differences between the pelvic fins of chimaeras and selachians. Chimaeras lack any radials that articulate with the pelvic girdle such as the propterygium found in selachian pelvic fins. However, despite this slight anatomical difference the development of the fins in *C. milli* does appear to corroborate that the basipterygium is derived from the fusion of radii in early development. The only other study of the development of the fin skeleton of *C. milli* has revealed, through the use of clearing and alcian blue staining, that in stage 29 embryos, mesenchymal rods are situated within the fin, and that the basipterygium is present by stage 31 and articulates laterally with these rods [41]. Using nanoCT imaging, we have visualised the development of the mesenchymal fin skeleton at stage 30 (Figure 3), which cannot be visualised with standard histological stains. Our findings demonstrate that the basipterygium forms from the proximal fusion of these “rods” supporting the hypothesis proposed by Dohrn [22] and Wiedersheim [21], and indicate that the pelvic fins of extant chimaeras and selachians occur through the same developmental mechanisms. The developmental origin of the pelvic girdle and its relationship with the fin radials during early development (prior to stage 30) remain unknown. Our data provide no evidence for Wiedersheim’s assertion that the pelvic girdle is also a result of the fusion of radii [21]. Further investigation using nanoCT imaging and specific condensing mesenchyme markers [64] on younger embryos of *C. milli* and other chondrichthyans should resolve these questions.

Despite similarities in the formation of the basipterygium, the timing of the segmentation of the pelvic skeleton differs between *C. milii* and *S. canicula*. In both taxa the pelvic girdle and basipterygium have chondrified prior to segmentation (Figure 6) [21], but segmentation begins in stage 32 in *C. milii* and stage 30 in *S. canicula*. The segmentation of the girdle and basipterygium in *S. canicula* is described as resulting from the formation of a “resorption zone” between the structures [21]. Resorption is not known to occur in chondrichthyans, which are believed to be incapable of repairing or remodelling their endoskeleton [65,66,67], but see [68,69,70]. Therefore, it is unlikely that resorption occurs in the subdivision forming these joints. The difference in alcian blue staining and pre-chondrogenic morphology of the tissue in the developing joints in *C. milii* at stage 32 (Figure 6) may indicate that joint formation/segmentation is the result of the repression of chondrogenesis as has been suggested in some tetrapods [71]. Our findings suggest that the fin joints of chimaeras are synarthotic, being separated by cartilaginous tissue as with the fin joints of some elasmobranchs [72,73].

To our knowledge the morphology of pelvic clasper skeletons has only been examined in *C. milii* [41] and not in any other chondrichthyan. Through clearing and alcian blue staining the earliest observation of the anterior and posterior clasper cartilages in *C. milii* is stage 33 [41]. In contrast, in the present study the formation of the pelvic claspers, both the primary anterior and posterior cartilages, was detected at stage 31 using standard histological staining (Figure 5F–H, AC, PC) as diffuse extensions from the condensation of the basipterygium. Despite this slight difference, which is likely a result of the different methods employed, the rest of the development of the pelvic claspers observed by clearing and staining agrees with our own with respect to the development of the clasper skeleton, including not observing the secondary components of the anterior clasper cartilage [41]. As the secondary components of the anterior clasper cartilage have not been observed embryonically it is likely these form and develop post hatching (i.e., after stage 36). Our findings concur with observations of clasper development based on external morphology in another chimaera, *H. colliei* [74], in which claspers can be identified by stage 31. Conversely, in a separate study of the external development of *C. milli*, claspers were first observed at stage 35 [46]. The morphology of clasper development in other chimaeroids and other chondrichthyans remains unknown and requires further study. Future investigations should employ multiple methods to gain a more comprehensive view of the development of these structures.

Prior to the present study the development of pre-pelvic tenacula has only been examined externally in both *C. milli* [46] and *H. colliei* [74]. Rudiments of the tenacula were observed externally as “bulges anterior to the pelvic fin,” at stage 30 in *C. milli* [46] and stage 31 in *H. colliei* [74]. We also found that the rudiments of this structure are present by stage 31 (Figure 5), with the grappler being identifiable by stage 32, and all elements being identifiable by stage 36 (Figure 6E,F, TG). The slight differences between our results and those based on external morphology of *C. milii* [46] may be due to individual variation or the bulges observed in a stage 30 *C. milii* [46] may have been concentrations of mesenchymal tissue prior to the formation of the condensations of the tenacula components.

### 4.3. The Implications of Chimaeroid Fin Development on the Evolution of Chondrichthyan Fins

It has been suggested that over the course of the evolution of chondrichthyans a transition in the morphology occurs, in which the basipterygium elongates and the fin radial articulation shifts from the pelvic girdle to the basipterygium [35]. This transition is borne out in fossil record of both elasmobranchs and holocephalans. In basal stem holocephalans, such as Symmoriiformes [66,67,69,70,71,73], and stem elasmobranchs such as the Xenacanthiforms [59,60,75,76,77] (Figure 9) the majority of the fin radials articulate directly with girdle and possess a rudimentary basipterygium [7]. In contrast, in taxa from clades closer to the holocephalan and elasmobranch crown groups, such as the Chondrenchelyiformes [53,78,79], Iniopterygiformes [54,80], and Hybodontiformes [57,58,60,61,81,82], the majority of the fin radials articulate directly with a relatively larger and longer basipterygium. In extant holocephalans and elasmobranchs the pelvic fin radials articulate solely or almost exclusively with the basipterygium [40]. Our developmental data support the hypothesis [21,22] that the basipterygium in extant chondrichthyans develops from the fusion of fin radials during early development, prior to chondrification. This suggests that the developmental mechanism responsible for the alteration of the size and location of the basipterygium and whence the area of fin radial articulation over the course of the evolution of chondrichthyans is changes in the region and degree of radial fusion during early development. Remarkably, this developmental mechanism is likely an example of parallel evolution [83], as the fossil record of both the Elasmobranchii [58,59,61,81,82] and Holocephali [40,50,51,54,56,79,80,84,85,86] would indicate that the resulting evolutionary transitions occurred independently (Figure 9).

## 5. Conclusions

Our findings indicate that a significant amount of patterning of the pelvic skeleton occurs during early development. During the early stages of the development of *C. milii*, the fin radials articulate directly on the pelvic girdle prior to fusing to form the basipterygium resembling the arrangement of both stem elasmobranchs, stem holocephalans, and placoderms. These findings corroborate the observations in historical descriptions of the development of *Scyliorhinid* sharks and suggest that chimaeras and selachians have similar developmental patterning in the formation of the pelvic girdle and fin. The formation of the basipterygium from the fusion fin radials in both extant holocephalans and elasmobranchs indicates that alterations in radial fusion during early development may explain alterations in the size and location of the basipterygium and regions of radial articulation over the course of the evolution of elasmobranchs and holocephalans. Further, the fossil record of the two chondrichthyan clades indicates that these evolutionary changes occurred independently in parallel. Our approach examining development through a combination of nanoCT imaging and histology has facilitated observation of the very early development of the skeleton and should be widely adopted in future studies of skeletal development. Combining this developmental approach with critical examination of the fossil record may help elucidate other details about the evolution of the appendicular skeleton in jawed vertebrates.

## Figures and Tables

**Figure 1 jdb-10-00053-f001:**
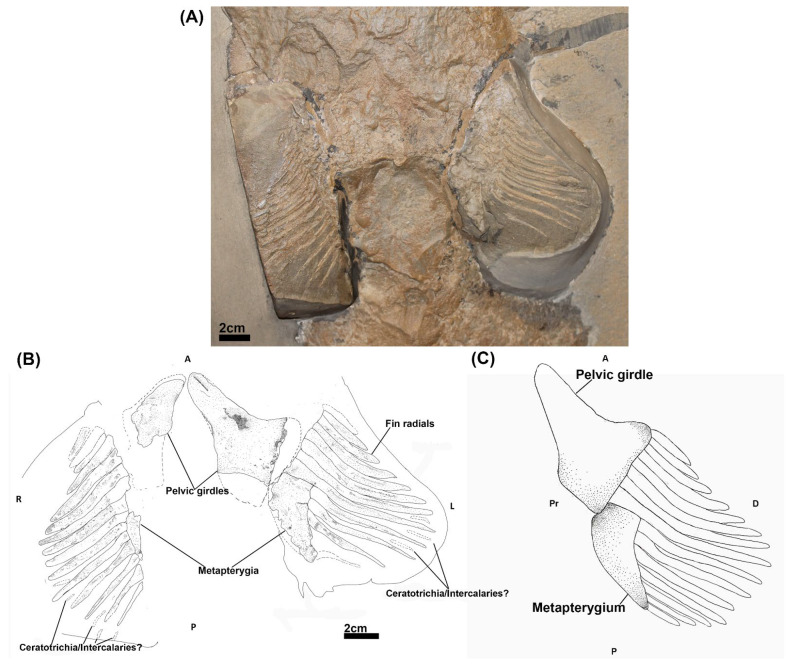
The pelvic skeleton of *Cladoselache kepleri* (NHMUK PV P 9269). (**A**) Photograph of the pelvic region of *C. kepleri* (NHMUK PV P 9269); (**B**) line drawing of *C. kepleri* (NHMUK PV P 9269), with potential extent of preserved elements indicated by dashed lines; (**C**) reconstruction of *C. kepleri* pelvic skeleton. Anatomical orientation indicated by A-anterior D-distal P-posterior Pr-proximal R-right L-left V-ventral.

**Figure 2 jdb-10-00053-f002:**
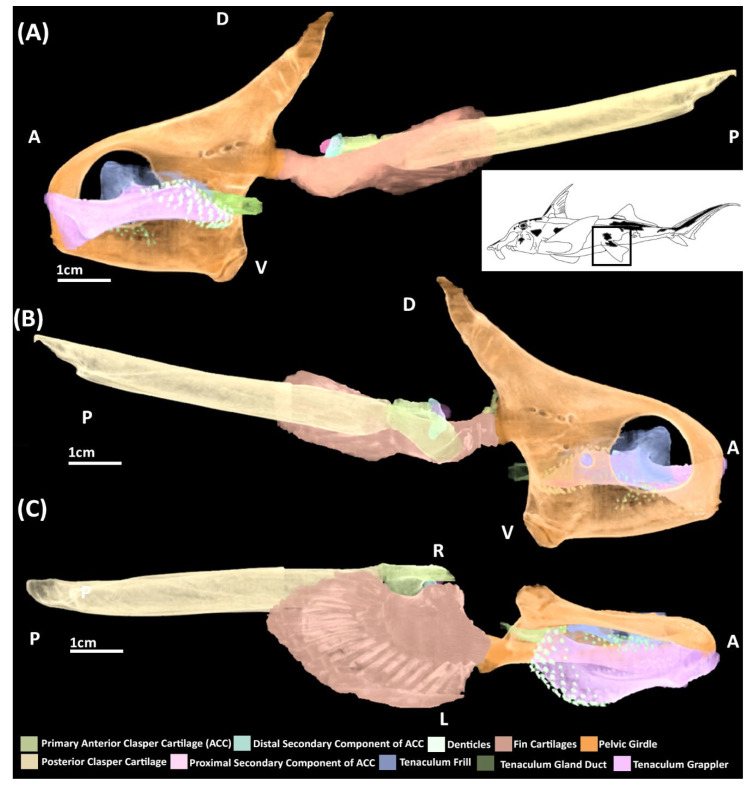
MicroCT models of the pelvic skeleton of an adult male elephant shark (*Callorhinchus milii*). (**A**) Models viewed from a distal lateral view; (**B**) models viewed from a proximal lateral view; (**C**) models viewed from a ventral view.

**Figure 3 jdb-10-00053-f003:**
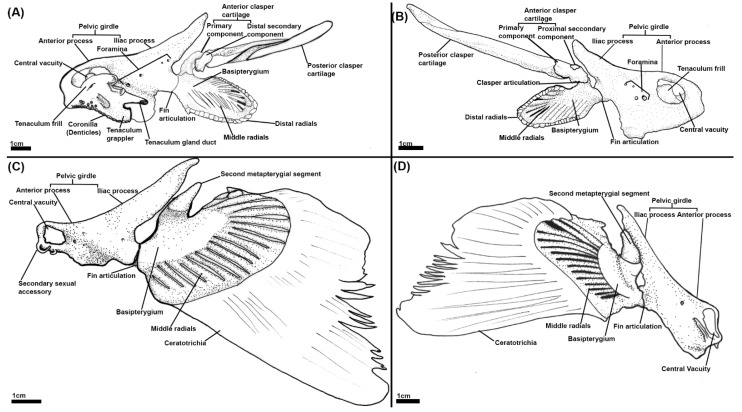
Illustrations of the pelvic skeleton of male and female elephant sharks (*Callorhinchus milii*) (**A**–**D**): (**A**) Distal lateral view of a male *C. milli* pelvic skeleton; (**B**) proximal lateral view of a male *C. milli* skeleton; (**C**) distal lateral view of a female *C. milli* pelvic skeleton; (**D**) proximal lateral view of a female *C. milli* pelvic skeleton.

**Figure 4 jdb-10-00053-f004:**
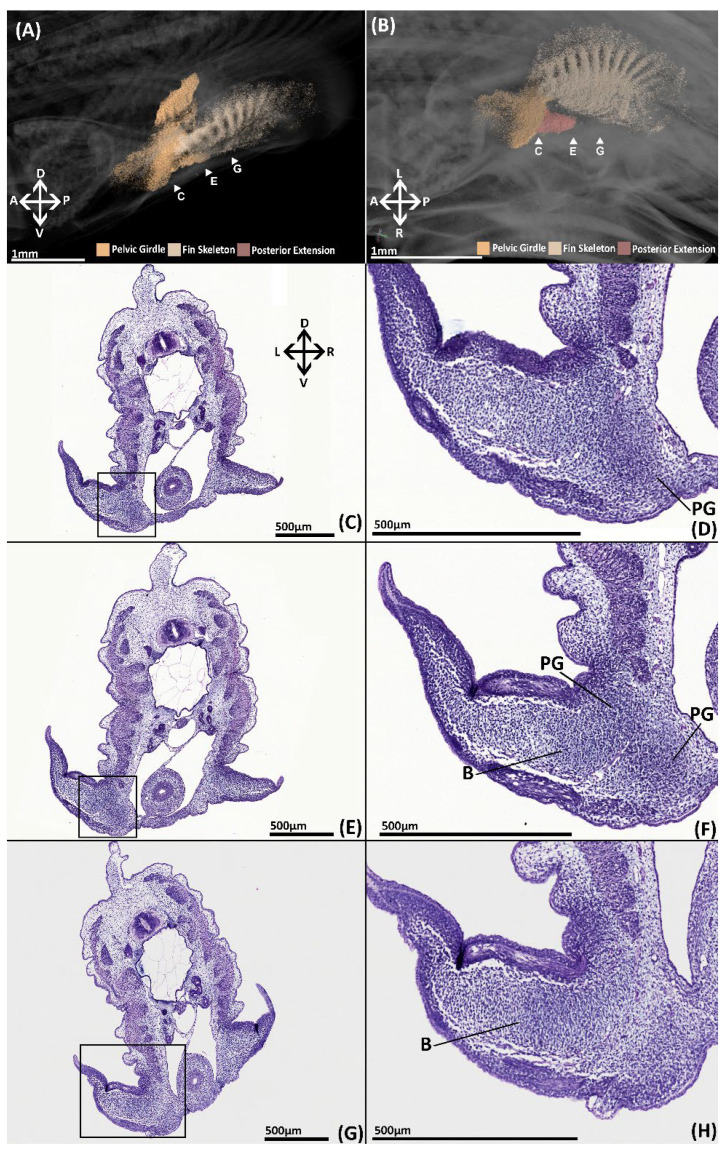
NanoCT models (**A**,**B**) and transverse paraffin sections (**C**–**H**) of a male stage 30 elephant shark (*Callorhinchus milii*) embryos. Sections are stained with haematoxylin, eosin and alcian blue and presented in an anteroposterior order. (**A**) 3D models of the pelvic skeleton of *C. milli* from a distal lateral view showing the span of the pelvic girdle; (**B**) same models from a ventral view showing the rays forming the basipterygium and the articulation of the fin skeleton on the pelvic girdle; (**C**) trunk section showing early mesenchymal condensation of the pelvic girdle; (**D**) close up of C within the black box; (**E**) trunk section showing the early mesenchymal pelvic and basipterygium; (**F**) close up of E within the black box; (**G**) trunk section showing extent of the mesenchymal basipterygium within the pelvic fin; (**H**) close up of G within the black box. B-basipterygium PG-pelvic girdle. Anatomical orientation of models indicated by A-anterior D-dorsal P-posterior R-right L-left V-ventral.

**Figure 5 jdb-10-00053-f005:**
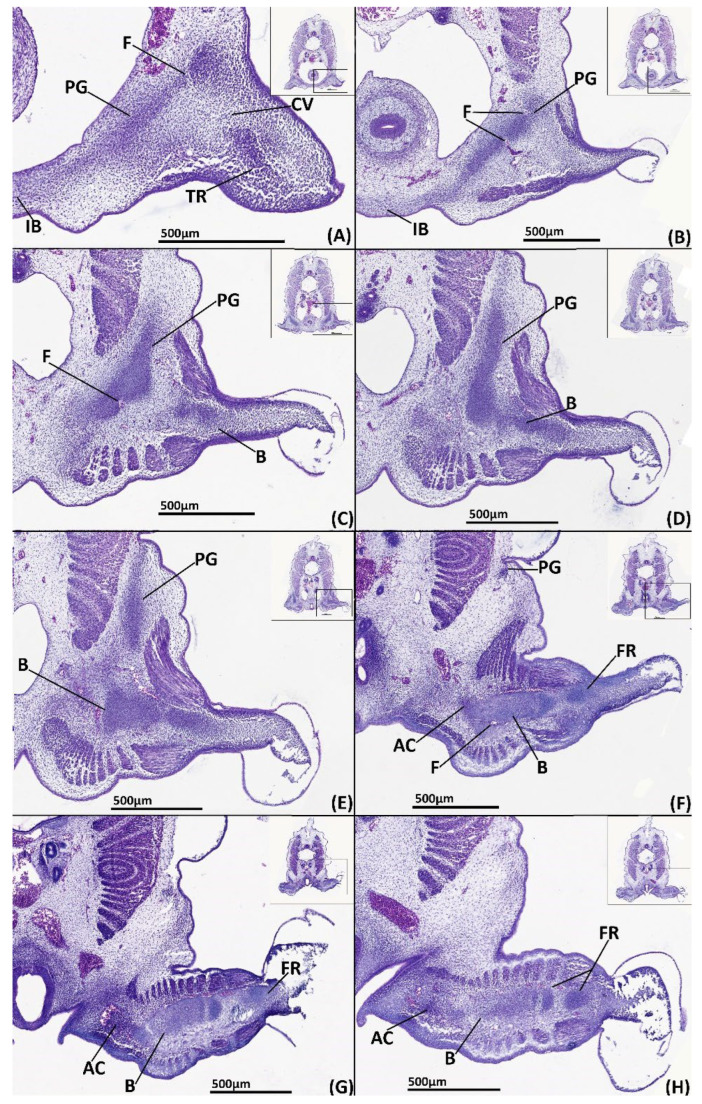
Close ups of transverse paraffin sections of a male stage 31 elephant shark (*Callorhinchus milii*) embryo stained with haematoxylin, eosin and alcian blue, presented in an anteroposterior order. The position of each close up is indicated by black boxes in the inset images. (**A**) close up of trunk section showing the anterior process of the mesenchymal pelvic girdle, the central vacuity and the tenacula rudiments; (**B**) close up of trunk section showing the posterior region of the anterior process and the first foramina anterior of the iliac process; (**C**) close up of trunk section showing the beginning of the mesenchymal iliac process and anterior region of basipterygium; (**D**) close up of trunk section showing the connection mesenchymal pelvic girdle and basipterygium; (**E**) close up of trunk section showing the mesenchymal basipterygium and dorsal regions of the mesenchymal iliac process; (**F**) close up of trunk section showing the connection of the mesenchymal basipterygium with a fin radial and the mesenchymal anterior clasper cartilage; (**G**) close up of trunk section showing the connection of the mesenchymal basipterygium and anterior clasper cartilage; (**H**) close up of trunk section showing the connection between the posterior regions of the basipterygium and the anterior clasper cartilage.AC-anterior clasper cartilage B-basipterygium CV-central vacuity F-foramina FR-fin radial IB-inter-pelvic band PC-posterior clasper cartilage PG-pelvic girdle. Anatomical orientation of models indicated by A-anterior D-dorsal P-posterior R-right L-left V-ventral.

**Figure 6 jdb-10-00053-f006:**
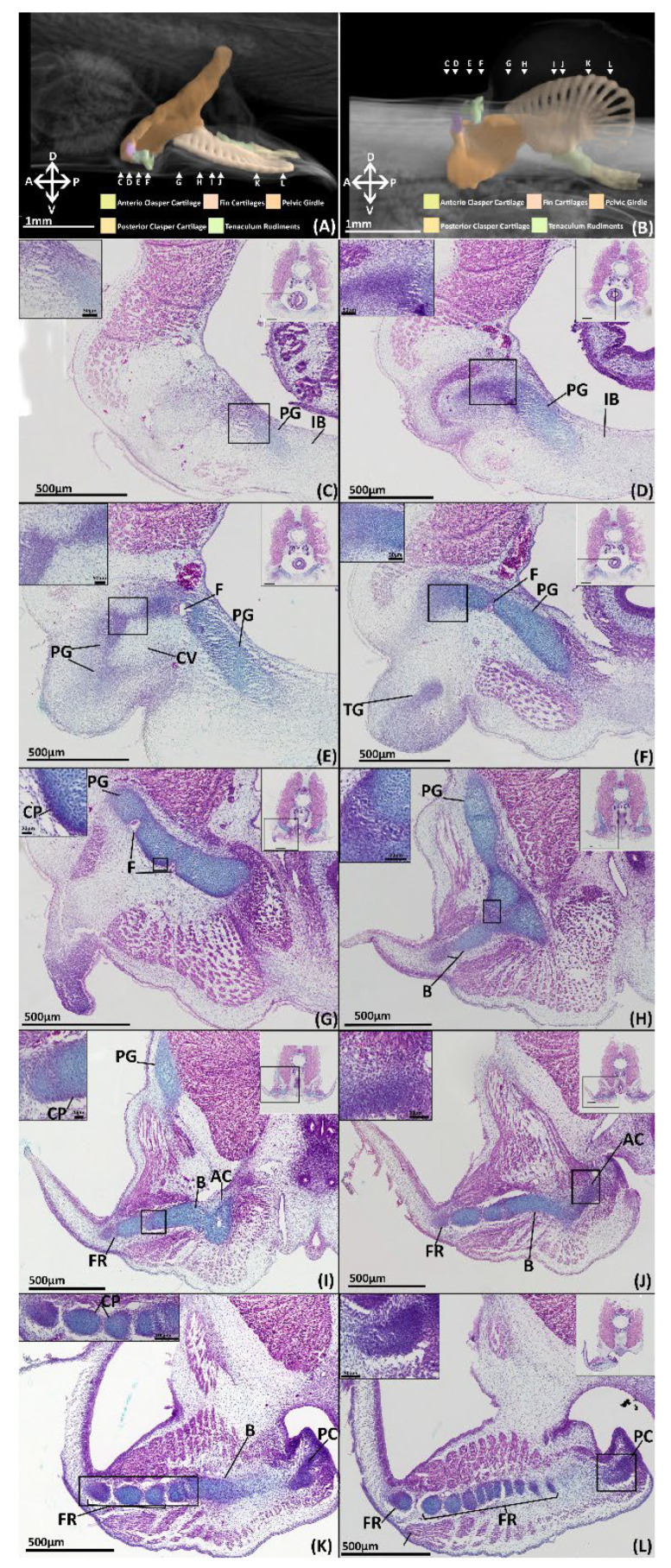
NanoCT models (**A**,**B**) and transverse paraffin sections (**C**–**L**) of a male stage 32 elephant shark (*Callorhinchus milii*) embryo. Paraffin sections are stained with haematoxylin, eosin and alcian blue and presented in an anteroposterior order. The position of each close up is indicated by black boxes in the inset images. (**A**) Models of the pelvic skeleton of *C. milli* from a distal lateral view; (**B**) same models from a ventral view; (**C**) close up of trunk section showing the anterior most point of the anterior process of the chondrogenic pelvic girdle; (**D**) close up of trunk section showing the anterior process of the chondrogenic pelvic girdle; (**E**) close up of trunk section showing anterior process of the pelvic girdle and the mesenchymal rudiments of the tenaculum; (**F**) close up of trunk section showing the mesenchymal rudiment of the tenaculum grappler; (**G**) close up of trunk section showing the lower region of the iliac process; (**H**) close up of trunk section showing the segmentation of the pelvic girdle and basipterygium; (**I**) close up of trunk section showing the dorsal regions of the pelvic girdle and the connection of the basipterygium with the anterior clasper cartilage and a fin radial; (**J**) close up of trunk section showing the mesenchymal anterior clasper cartilage and its connection with the basipterygium; (**K**) close up of trunk section showing the posterior regions of the fin skeleton and the mesenchymal posterior clasper cartilage; (**L**) close up of trunk section showing the fin radials and mesenchymal posterior clasper cartilage. AC-anterior clasper cartilage B-basipterygium CP-perichondrium CV-central vacuity F-foramina FR-fin radial(s) PC-posterior clasper cartilage PG-pelvic girdle TG-tenaculum grappler. Anatomical orientation of models indicated by A-anterior D-dorsal P-posterior R-right L-left V-ventral.

**Figure 7 jdb-10-00053-f007:**
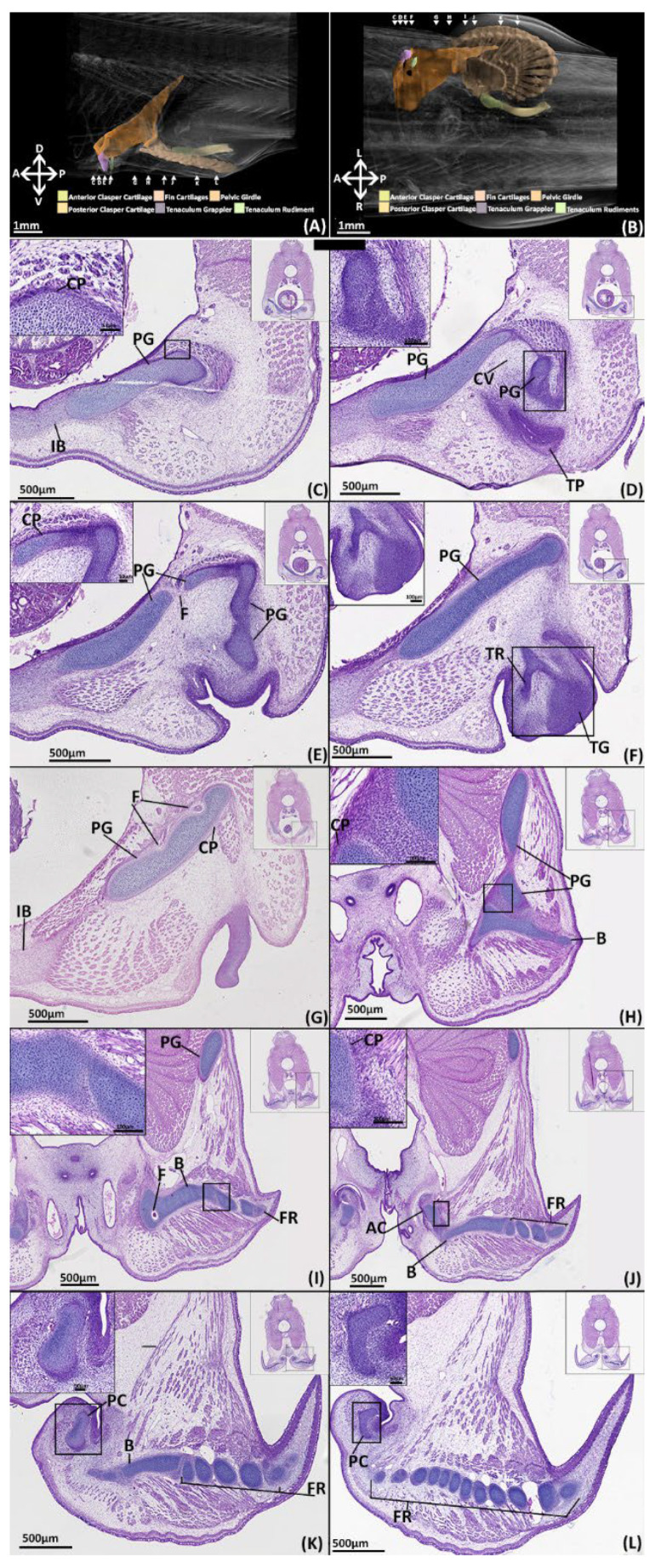
NanoCT models (**A**,**B**) and transverse paraffin sections (**C**–**L**) of a male stage 34 elephant shark (*Callorhinchus milii*) embryo. Transverse paraffin sections are stained with haematoxylin, eosin and alcian blue and presented in an anteroposterior order. The position of each close up is indicated by black boxes in the inset images. (**A**) models of the pelvic skeleton of *C. milli* from a distal lateral view; (**B**) models from a dorsal view; (**C**) close up of a trunk section showing the anterior region of the pelvic girdle; (**D**–**E**) close up of a trunk section showing the anterior process of the pelvic girdle and the condensing of the cartilages framing the central vacuity; (**F**) close up of a trunk section showing the anterior process and the condensing tenaculum grappler; (**G**) close up of a trunk section showing the lower dorsal region of the pelvic girdle; (**H**) close up of a trunk section showing the point of articulation between the pelvic girdle and the basipterygium; (**I**) close up of a trunk section showing the dorsal region of the pelvic girdle and the basipterygium and its connection with some fin radials; (**J**) close up of a trunk section showing the articulation between the chondrogenic anterior clasper cartilage and the basipterygium; (**K**) close up of a trunk section showing the anterior region of the posterior clasper cartilage and posterior regions of the basipterygium; (**L**) close up of a trunk section showing posterior regions of the posterior clasper cartilage and fin radials. AC-anterior clasper cartilage B-basipterygium CP-perichondrium CV-central vacuity F-foramina FR-fin radial(s) PC-posterior clasper cartilage PG-pelvic girdle TG-tenaculum grappler TR-tenacula rudiments. Anatomical orientation of models indicated by A-anterior D-dorsal P-posterior R-right L-left V-ventral.

**Figure 8 jdb-10-00053-f008:**
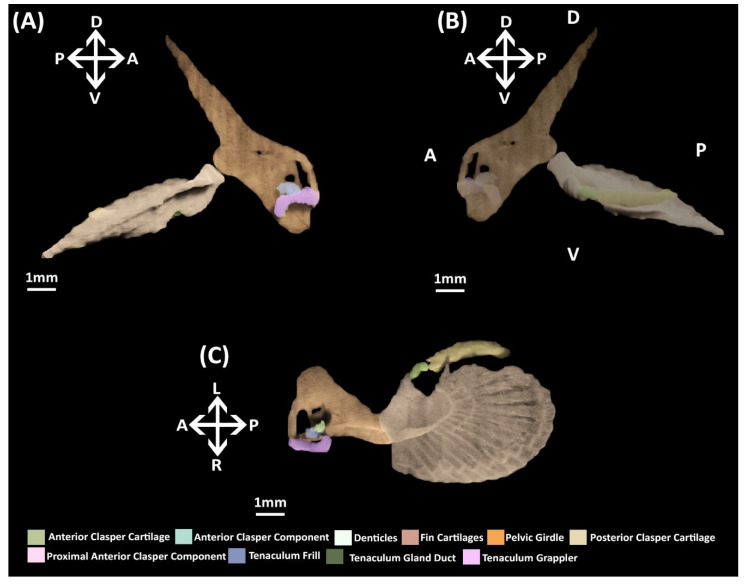
NanoCT models of a male stage 36 elephant shark (*Callorhinchus milii*) embryo. (**A**) Models of the pelvic skeleton from a distal lateral view; (**B**) same models from a proximal lateral view; (**C**) models from a ventral view. Anatomical orientation of models indicated by A-anterior D-dorsal P-posterior R-right L-left V-ventral.

**Figure 9 jdb-10-00053-f009:**
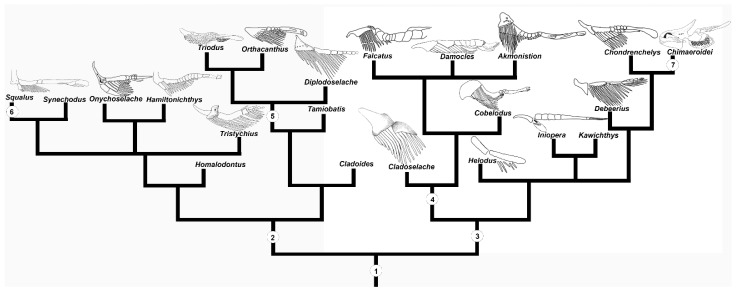
Chondrichthyan phylogeny (after [8]) with reconstructions of the pelvic skeletons. 1- Chondrichthyes 2- Elasmobranchii 3- Holocephali 4- Symmoriiformes 5- Xenacanthiforms 6- Crown group Elasmobranchii 7- Crown group Holocephali. Illustrations redrawn from [1,2,49,50,51,52,53,54,55,56,57,58,59,60,61,62].

**Figure 10 jdb-10-00053-f010:**
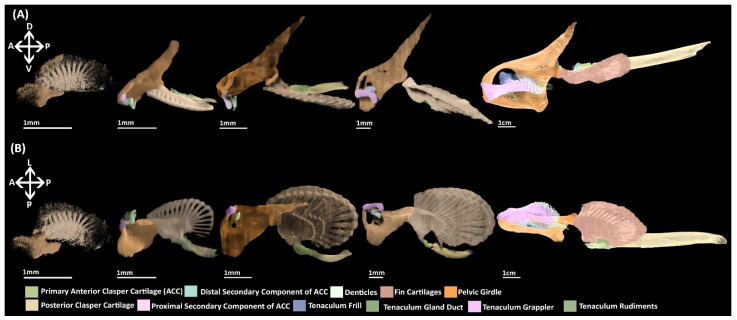
NanoCT models of the development of the pelvic skeleton in *C. milii*. (**A**) Models observed in a distal lateral view; (**B**) Models in a ventral view. Anatomical orientation of models indicated by A-anterior D-dorsal P-posterior R-right L-left V-ventral.

**Table 1 jdb-10-00053-t001:** Staining duration and imaging parameters of specimens.

Stage	Staining (days)	Voltage (kV)	Power (W)	Exp. Time (sec)	Scan Time (hr)	Voxel Size (μm)
30	27	60	5	3.0	5.0	3.0
32	67	60	5	4.5	6.5	4.0
34	69	60	5	5.0	7.0	5.5
36	77	80	7	2.0	9.0	12.0
Adult	91	65	25	1.9	39.1	8.7

## Data Availability

The data presented in this study are available on request from the corresponding author. The data are not publicly available due to ongoing research.

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
