# Peer review of "The Development of the *Chimaeroid* Pelvic Skeleton and the Evolution of Chondrichthyan Pelvic Fins"

_jdb, 2022, doi:10.3390/jdb10040053_

Round 1

Reviewer 1 Report

Punctuation, grammar, accurate positional descriptions, figure citations, - check throughout.

Placoderms and acanthodians in present context are grades not clades (remember, clade= branch = natural group).

General comment – improves towards the end, but paper starts poorly – worth persisting.

Establish terminology at the start of the descriptions and stick to it – be consistent throughout the article.

Reduce text where possible and rely on illustrations. Add illustrations if necessary.

L50-57. “Chondrichthyes have long been used as models to examine the evolution and development of the appendicular anatomy [13-15], often under the false notion that they are primitive, and that their morphology represented the “ancestral” body plan [13, 14, 16, 54 17]. Chondrichthyans instead possess a mixture of derived and plesiomorphic morphologies which can be examined and used to better understand the evolution of forms across the gnathostome clade when compared with extant and fossil taxa [30].”

First statement – fine, but response is vague with an odd citation of [30] - Bendix-Almgreen’s description of a Cladoselache pectoral fin.

L75-76. A more constructive comment and comparison would be an improvement.

What extra information is supplied here, relative to the material delivered by Riley et al. 2017?

How does the current manuscript build on this recent description, and how do the questions behind these comparable papers differ?

L156+. Photograph of specimen NHMUK PV P 9269 required – all previous mentions of this specimen are ~100yrs old.

For a comparative brief & effective report on a fossil fin, see Tomita’s 2015 (ref #29) description of pectoral fins published in JVP. Suggest using this as a model in your revision.

Cladoselache has a single pelvic girdle consisting of separate left and right plates; not two pelvic girdles.

Process present - not a point.

‘is a cone like metapterygial element which taper off’

Meaning a subtriangular metapterygial plate that tapers distally?

The fin is supported by at least eight pre-metapterygial radials; approximately seven radials articulate with the metapterygial plate.

What evidence for the ‘ceratotrichia’? Could these undescribed items be cartilage intercalaries, as found in the pectoral fins (see ref #30)?

What is the evidence for the line indicating ‘fin keel’ and ‘fissure metapterygii’?

L177+

Callorhinchus adult morphology – description and figure 2

Figure 2 is superficially attractive but fails in a couple of important respects.

The girdles and fins need to be enlaged: more visible morphology.

The images lack clarity. Because of the semi-transparent renderings the colours are insufficiently distinct, and sorting greens from oranges will be especially challenging for anyone with limited colour perception.

Terminology – what is the source of the term ‘grappler’?

I suggest sticking to more generally used terminology such as names employed in Ref 35 -Didier 1995. ‘Grappler’ implies knowledge of function (grappling): entertaining, but is there any citable source? Didier 1995 suggests none. Any new observations since ’95?

Figure 3 needs to be replaced.

The line drawings are inadequate, primarily because divisions and articulations between cartilages are missing. The depiction is limited to an approximate outline of large areas of cartilage skeleton. In part A, several items tagged are simply not shown, e.g., ‘Fin articulation’.

Labels are confusing and, in some instances, unfamiliar.

Once again, ‘Grappler’ rather than ‘pre-pelvic clasper or tenaculum’.

What is a ‘Secondary sexual accessory’? Accessory what?

What is a ‘Second fin metapterygium’?

See figures in Riley et al. 2017 (ref.38) for a reasonable standard of clear representation.

Somewhere around here, note that Didier et al. 1998, describe pelvic fins as just beginning to form at stage 27.

Figure 3 captions: what is a ‘lateral proximal’ or ‘lateral distal’ view?  These terms suggest proximity of viewer to lateral surface, but this interpretation doesn’t match the depictions.

L 202 ‘agglomeration’ -not sure this is the correct term. The pre-cartilaginous tissue shown in Fig. 4 is clearly distinguishable into rudiments of fin and girdle.

‘Iliac ramus’ – I advise strongly against using this term. There is no evidence that this process has any clear relationship with the iliac process of tetrapod pelves. Likewise, ‘obturator foramen’ – see Didier 1995 for a more careful use of terminology. This is not a tetrapod pelvis; this pattern is not ancestral to tetrapod pelves.

L208. ‘Proximally and mesially this tissue is forming between and along these rays forming the rudimentary basipterygium. ‘

What does this mean?

By ‘rays’, do the authors mean radials?

Are the radials forming the basipterygium, in a distal to proximal direction?

If so, how is this directionality known from a single snapshot?

‘very diffuse tissue at their distal margins (Figure3A-B) resembles the distal fin radials in the adult (Figure1, 2).’

Figure 1 shows an estimate of a Cladoselache pelvic skeleton. No distal radials.

Figure 2 shows MicroCT models of pelvic skeletons which lack sufficient detail to show distal radials, and none are labeled as such.

Figure 3 includes the label ‘distal radials’, but none is actually shown.

L215. A posterior extension of tissue can be observed projecting from the pelvic girdle (Figure3B).

Is the is red item in in Figure 4B?

Stage 31 description and Figure 5.

L292+ Claspers beginning to form – how is a reader to know that the area identified as ‘AC’ in 5F, G, and H is separate from area ‘B’ in parts 5C, D and E?

Readers need more than slices to understand the differentiation of these skeletal parts. Include figures of whole fins.

For each of these figures, add simple schematic of fin to show A-P level of slices.

Review the remainder of the description of embryonic stages and histological sections carefully for similar corrections/improvements.

Discussion and conclusion sections.

The discussion is generally good and the arguments are reasonable. However, a summary figure would improve the manuscript immensely, with a series of schematics demonstrating the authors’ interpretation of the sequential elaboration of the C. milli pelvic skeleton. This would aid readers’ use of the descriptive text and histology – ultimately making the work more citable.

Figure 9 – change the proportions to show more of the pelvic skeletons; don’t allow the branches of the tree to take up so much space.

Author Response

General response to the reviewers:

We thank the reviewers for their constructive comments and suggestions. The majority of the comments were typographical in nature or requests for minor clarification. Those have been addressed with track changes in the resubmitted version. A response to reviewer 1’s comments are below. Several comments were raised about the resolution of the figures presented in manuscript.  It should be noted that the figures contained in the manuscript are compressed and do not show the high resolution of the actual figures, which can be seen here https://www.dropbox.com/scl/fo/uv185f54pso4gh2k6r05s/h?dl=0&rlkey=6r7fgykdi8hb3epb8etgto1rm .

Response to Reviewer 1’s comments:

Comment: Punctuation, grammar, accurate positional descriptions, figure citations, - check throughout.

 Response: The manuscript has been revised to address these issues.

Comment: Placoderms and acanthodians in present context are grades not clades (remember, clade= branch = natural group).

Response: We have used the word groups in the introduction to reflect the debated nature of placoderm monophyly. In all other instances, appropriate phylogenetic nomenclature has been used.

Comment: General comment – improves towards the end, but paper starts poorly – worth persisting.

Response: In the introduction we have revised confusing wording raised by the reviewer, addressed a lack of engagement with some of the literature and clarified the scope and role of the different methodologies employed. We have addressed the typographical errors in the results and figure captions. The descriptions of the CT and histological data have been revised to make them more succinct where possible, by condensing some sentences and relying more on the figures to elaborate the findings.

Comment: Establish terminology at the start of the descriptions and stick to it – be consistent throughout the article.

Response: A terminology section has been added to the methods, line 161, citing the sources of the terminology used in the descriptions.

Comment: Reduce text where possible and rely on illustrations. Add illustrations if necessary.

Response: The text, particularly the results, has been made more succinct where possible.

Comment: L50-57. “Chondrichthyes have long been used as models to examine the evolution and development of the appendicular anatomy [13-15], often under the false notion that they are primitive, and that their morphology represented the “ancestral” body plan [13, 14, 16, 54 17]. Chondrichthyans instead possess a mixture of derived and plesiomorphic morphologies which can be examined and used to better understand the evolution of forms across the gnathostome clade when compared with extant and fossil taxa [30].”First statement – fine, but response is vague with an odd citation of [30] - Bendix-Almgreen’s description of a Cladoselache pectoral fin.

Response: The sentence has been amended for clarity: “Current understandings of the anatomy and fossil record of Chondrichthyans indicate that these taxa possess a mixture of derived and plesiomorphic morphologies which can be examined to investigate the evolution of different traits across the gnathostome clade when compared with extant and fossil taxa [4, 19, 20]." The original citation had lost its endnote formatting. The correct references, Trinajstic et al., 2018, Pradel et al. 2011 and Maisey et al. 2017, have now been cited.

Comment: L75-76. A more constructive comment and comparison would be an improvement.

What extra information is supplied here, relative to the material delivered by Riley et al. 2017?

How does the current manuscript build on this recent description, and how do the questions behind these comparable papers differ?

 Response: Riley et al. 2017 focus only on the development of paired fins and claspers and do not describe the development of the girdles. We have added this information to the introduction (lines 75) as requested. The findings of Riley et al. are already compared with our own in the second section of the discussion (line 524-539; 578-592; 607-625).

Comment: L156+. Photograph of specimen NHMUK PV P 9269 required – all previous mentions of this specimen are ~100yrs old.

Response: Figure 1 has been edited to provide a specimen photograph as requested.

Comment: For a comparative brief & effective report on a fossil fin, see Tomita’s 2015 (ref #29) description of pectoral fins published in JVP. Suggest using this as a model in your revision.

Response: Figure 1 has been edited accordingly.

Comment: Cladoselache has a single pelvic girdle consisting of separate left and right plates; not two pelvic girdles.

Process present - not a point.

 Response: We agree with the reviewer that there are two distinct and separated girdles (left and right) and as they are not connected by a symphysis, the use of “girdles” is appropriate. Separate paired pelvic girdles are considered the plesiomorphic condition of the Chondrichthyes as per Coates, 2003 and Lund & Grogan, 1997. “Point/s” has been substituted with “process/es”.

Comment: ‘is a cone like metapterygial element which taper off’

Meaning a subtriangular metapterygial plate that tapers distally?

Response: The description of the shape of the metapterygium has been amended accordingly, see results section 3.1. These elements taper posteriorly not distally as per the text.

Comment: The fin is supported by at least eight pre-metapterygial radials; approximately seven radials articulate with the metapterygial plate.

Response: As per figure 1, the relationship between the radials and the adjacent girdles are uncertain due to poor preservation and damage of the specimen. The interpretation of their arrangement is therefore given in the discussion not the results.

Comment: What evidence for the ‘ceratotrichia’? Could these undescribed items be cartilage intercalaries, as found in the pectoral fins (see ref #30)?

Response: The presence of intercalated radials in Cladoselache is controversial. Bendix-Almgreen considered these to be preservation artifacts resulting from the splits in the heavy calcification of the fin radials. Tomita (2015) does not address this issue, appearing to follow Dean (1909). We have amended the description (line 190) and Figure 1 to consider both possibilities, as we do not have any reason to dispute either argument.

Comment: What is the evidence for the line indicating ‘fin keel’ and ‘fissure metapterygii’?

 Response: The figure was initially labelled following Bendix-Almgreen’s (1975) interpretations of the pelvic fin anatomy. These labels have been removed from Figure 1 as we are no longer certain of the identity of these structures, and they are not relevant to the scope of the paper.  

 Comment: L177+ Callorhinchus adult morphology – description and figure 2

 Figure 2 is superficially attractive but fails in a couple of important respects.

The girdles and fins need to be enlaged: more visible morphology. The images lack clarity. Because of the semi-transparent renderings the colours are insufficiently distinct, and sorting greens from oranges will be especially challenging for anyone with limited colour perception.

Response: The models have been enlarged for clarity. As for the transparent renderings, they have been designed to show smaller components of the anterior clasper cartilage that would be hidden in opaque images. We acknowledge the reviewers concerns regarding colour blindness. We have examined the figure in photoshop using different colour blindness filters and have found that the elements can still be distinguished from each other.

Comment: Terminology – what is the source of the term ‘grappler’?

Response: Grappler is used sensu Leigh-Sharpe (1922), who divides the tenacula into three distinct components, the grappler, the frill and gland duct. Didier (1995) does describe these components but provides no terminology. A terminology section has been added to the methods to clarify the sources for the terminologies used in the descriptions.

Comment: I suggest sticking to more generally used terminology such as names employed in Ref 35 -Didier 1995. ‘Grappler’ implies knowledge of function (grappling): entertaining, but is there any citable source? Didier 1995 suggests none. Any new observations since ’95?

 Response: As per the previous response we use the terminology of Leigh-Sharpe (1922) to describe the components of the tenaculum. We also note that grappler implies as much function as tenaculum, both are implements to hold or seize objects. Contra Didier (1995), Jones et al. (2005) argue that the tenacula insert into the pre-pelvic slits of females during copulation. See https://www.taylorfrancis.com/chapters/edit/10.1201/9781439856000-27/male-genital-ducts-copulatory-appendages-chondrichthyans-carolyn-jones-terence-walker-jus-tin-bell-matt-reardon-carlos-ambrosio-adriana-almeida-william-hamlett. Based on the mating scars observed on females, it is more probable that the grappler is used to stabilize the male on the flank of the females to insert the clasper in the cloaca.

Comment: Figure 3 needs to be replaced.

The line drawings are inadequate, primarily because divisions and articulations between cartilages are missing. The depiction is limited to an approximate outline of large areas of cartilage skeleton. In part A, several items tagged are simply not shown, e.g., ‘Fin articulation’.

Labels are confusing and, in some instances, unfamiliar.

Once again, ‘Grappler’ rather than ‘pre-pelvic clasper or tenaculum’.

What is a ‘Secondary sexual accessory’? Accessory what?

What is a ‘Second fin metapterygium’?

See figures in Riley et al. 2017 (ref.38) for a reasonable standard of clear representation.

 Response: Figure 3 has been edited to clarify these details and the labels have been revised as per the previous response. The secondary sexual accessory is the female equivalent of the pre-pelvic tenaculum according to Didier (1995). The stippling indicates relief as per standard morphological and palaeontological conventions. The missing articulations which have now been corrected.

Comment: Somewhere around here, note that Didier et al. 1998, describe pelvic fins as just beginning to form at stage 27.

 Response: The developmental staging table published by Didier et al 1998 uses external morphology for staging and only refers to the fin buds at this stage. Whilst it is the staging table that has been used for this work and is extremely helpful, the presence of fin buds does not necessarily indicate the presence of skeletal components and the techniques we have used in this study are necessary to establish that.

Comment: Figure 3 captions: what is a ‘lateral proximal’ or ‘lateral distal’ view?  These terms suggest proximity of viewer to lateral surface, but this interpretation doesn’t match the depictions.

Response: The caption has been revised to clarify the views depicted.

Comment: L 202 ‘agglomeration’ -not sure this is the correct term. The pre-cartilaginous tissue shown in Fig. 4 is clearly distinguishable into rudiments of fin and girdle.

Response: The term “agglomeration” has been replaced with “condensation” in line 228 and subsequent. The rudiments of different skeletal components can be identified figure 4 a and b.

Comment: ‘Iliac ramus’ – I advise strongly against using this term. There is no evidence that this process has any clear relationship with the iliac process of tetrapod pelves. Likewise, ‘obturator foramen’ – see Didier 1995 for a more careful use of terminology. This is not a tetrapod pelvis; this pattern is not ancestral to tetrapod pelves.

 Response: The use of these terms was not intended to suggest any homology between these structures and those of the pelvises of tetrapods. We have substituted these terms with iliac process and central vacuity (sensu Didier, 1995) line 231 and subsequent.

Comment: L208. ‘Proximally and mesially this tissue is forming between and along these rays forming the rudimentary basipterygium. ‘

What does this mean?

By ‘rays’, do the authors mean radials?

Are the radials forming the basipterygium, in a distal to proximal direction?

If so, how is this directionality known from a single snapshot?

Response: The description of the stage 30 CT models has been revised to clarify these points see discussion section 3.2.1.1.

Comment: ‘very diffuse tissue at their distal margins (Figure3A-B) resembles the distal fin radials in the adult (Figure1, 2).’

Figure 1 shows an estimate of a Cladoselache pelvic skeleton. No distal radials.

Figure 2 shows MicroCT models of pelvic skeletons which lack sufficient detail to show distal radials, and none are labeled as such.

Figure 3 includes the label ‘distal radials’, but none is actually shown.

Response: This was a typographical error and has been corrected. Figure 3 has been revised to show the distal fin radials. The distal radials are visible in Figure 2C.

Comment: L215. A posterior extension of tissue can be observed projecting from the pelvic girdle (Figure3B).

Is the is red item in in Figure 4B?

 Response: The figure citation has been corrected. The red structure is the posterior extension in Figure 4b. The legend has been enlarged for clarity.

Comment: Stage 31 description and Figure 5.

L292+ Claspers beginning to form – how is a reader to know that the area identified as ‘AC’ in 5F, G, and H is separate from area ‘B’ in parts 5C, D and E?

Response: The description has been revised to clarify that the anterior clasper cartilage is an extension of the basipterygium at this stage of development (line273 onwards).

Comment: Readers need more than slices to understand the differentiation of these skeletal parts. Include figures of whole fins.

Response: The histological data is presented to understand the cellular aspects of development and growth, whilst the CT data is presented to understand the morphology of the skeleton including the fins. This has been clarified in the methods. We do not possess scan data of stage 31 and cannot provide images of the whole fins for this stage. The significance of this stage is to illustrate that the pelvic girdle and the basipterygium are still continuous as a now histologically visible condensation. Components of the fin skeleton can be distinguished using transverse sections as evidenced by the labelling of the components in the slices.

Comment: each of these figures, add simple schematic of fin to show A-P level of slices.

 Response: The approximate location of the slices is now indicated on the CT models for stages 30, 32, and 34 with arrowheads.

Comment: Review the remainder of the description of embryonic stages and histological sections carefully for similar corrections/improvements.

 Response: The descriptions, particularly those of the histological data, have been revised to make the text more succinct where possible. Any typographical errors and unrevised figure citations have been corrected.

Comment: Discussion and conclusion sections.

The discussion is generally good and the arguments are reasonable. However, a summary figure would improve the manuscript immensely, with a series of schematics demonstrating the authors’ interpretation of the sequential elaboration of the C. milli pelvic skeleton. This would aid readers’ use of the descriptive text and histology – ultimately making the work more citable.

Response: A summary figure of the CT models in lateral and ventral views has been added to the second section of the discussion (Figure 10) for a convenient overview of the development of the pelvic skeleton.

Comment: Figure 9 – change the proportions to show more of the pelvic skeletons; don’t allow the branches of the tree to take up so much space.

Response: The proportions of the pelvic skeletons have been increased for clarity.

Reviewer 2 Report

In this article, Pears and colleagues combine traditional methods such as dissections and histology to nanoCT imaging to evaluate the adult and the developing pelvic fins at stages 30,31,32 and 34, of the elephant sharkC. milii. To obtain a better comparative evolutionary perspective, they also revaluate the pelvic fin of Cladoselache, a fossil stem holocephalan. Their findings are ainterpreted within the context of pelvic fin morphology (and development) across Chondrichthyes. 

This is a well written, the conclusions are sound, detailed description of fin development, the nanoCT images striking and the data will be of great value to comparative anatomists interested in the development, function and evolution of the pelvic skeleton in Chondrichthyes and in vertebrates in general.

I believe the written manuscript and data are of high quality in their current state. I have only minor, mostly typographical errors, to provide as feedback. I congratulate the authors on their excellent manuscript.

Pg1 Ln38: remove period.

Pg3 Ln93: “Whole body…” sentence is incomplete

Pg4 Ln94: “ This specimen.. “ remove “of”

Pg4 Ln107. Citation is spelled out. Should be a number instead.

Pg4 Ln119: “in” is duplicated

Pg11 Ln303: Should read” shorter than that of…”

Pg14 Ln 389 and 395: Authors refer to Figure5 on both lines. However, I believe they meant to refer to Figure6

Pg18 Ln535: Study (instead of studies)

Pg19 Ln560: in (instead of int)

Pg20 ln612: missing period after “basipterygium”.

Pg 20 Ln618: “Our findings concur with…” Revise sentence please.

Author Response

General response to the reviewers:

We thank the reviewers for their constructive comments and suggestions. The majority of the comments were typographical in nature or requests for minor clarification. Those have been addressed with track changes in the resubmitted version. A response to reviewer 1’s comments are below. Several comments were raised about the resolution of the figures presented in manuscript.  It should be noted that the figures contained in the manuscript are compressed and do not show the high resolution of the actual figures, which can be seen here https://www.dropbox.com/scl/fo/uv185f54pso4gh2k6r05s/h?dl=0&rlkey=6r7fgykdi8hb3epb8etgto1rm .

Pg1 Ln38: remove period.

Pg3 Ln93: “Whole body…” sentence is incomplete

Pg4 Ln94: “ This specimen.. “ remove “of”

Pg4 Ln107. Citation is spelled out. Should be a number instead.

Pg4 Ln119: “in” is duplicated

Pg11 Ln303: Should read” shorter than that of…”

Pg14 Ln 389 and 395: Authors refer to Figure5 on both lines. However, I believe they meant to refer to Figure6

Pg18 Ln535: Study (instead of studies)

Pg19 Ln560: in (instead of int)

Pg20 ln612: missing period after “basipterygium”.

Pg 20 Ln618: “Our findings concur with…” Revise sentence please.

Response: The typographical errors have been corrected accordingly and the sentence at the end of the second section of the discussion has been revised.

Reviewer 3 Report

This manuscript describes development of the pelvic fin skeleton in the elephant shark using sophisticated imaging methods. The level of detail in the images is extraordinary. The results suggest a potential developmental mechanism that may have produced morphological changes in the pelvic fin over chondrichthyan evolution. Specifically, changes in the fusion of fin radials during early development could produce the different fin configurations that exist in this clade. The manuscript is very well written and logically organized. Rationale and goals are clear from the beginning, and the figures are high quality and clearly labeled. I have just a few questions and suggestions for clarity.

Figure 2: maybe add an inset with an image of the whole shark showing the location and orientation of the pelvic fin for context.

Figure 3. Text is very small

Figure 4: Similarly, an image of the shark showing the location of the sections would be helpful

Figure 6. The nano-CT images here are absolutely beautiful but too small to see clearly. They also need to be referenced in the figure caption and the color scheme given. 

Figure 8: This is a nice, large image but the quality is not as good as the earlier stages. Why was this stage chosen for the full size image?

Figure 9: the images of fins are too small to see clearly. if this is the final figure size please reorganize the figure so that the images can be larger.

Lines 93-94: missing words?

Line 640: is "from" missing between shift and the?

Lines 652-656: This sentence is hard to read. Please shorten/reword

Line 662: in elephant sharks or a broader group?

Author Response

We thank the reviewers for their constructive comments and suggestions. The majority of the comments were typographical in nature or requests for minor clarification. Those have been addressed with track changes in the resubmitted version. A response to reviewer 1’s comments are below. Several comments were raised about the resolution of the figures presented in manuscript.  It should be noted that the figures contained in the manuscript are compressed and do not show the high resolution of the actual figures, which can be seen here https://www.dropbox.com/scl/fo/uv185f54pso4gh2k6r05s/h?dl=0&rlkey=6r7fgykdi8hb3epb8etgto1rm .

Comments and Suggestions for Authors

This manuscript describes development of the pelvic fin skeleton in the elephant shark using sophisticated imaging methods. The level of detail in the images is extraordinary. The results suggest a potential developmental mechanism that may have produced morphological changes in the pelvic fin over chondrichthyan evolution. Specifically, changes in the fusion of fin radials during early development could produce the different fin configurations that exist in this clade. The manuscript is very well written and logically organized. Rationale and goals are clear from the beginning, and the figures are high quality and clearly labelled. I have just a few questions and suggestions for clarity.

Response: The suggestions were helpful and constructive. The issues regarding the clarity and details of the figures primarily arise from compression of the images inserted in the word/pdf document. The actual figures are of a high resolution in which the details can be seen clearly.

Comment: Figure 2: maybe add an inset with an image of the whole shark showing the location and orientation of the pelvic fin for context.

Response: An inset image of a whole shark has been added to show the pelvic region in Figure 2.

Comment: Figure 3. Text is very small

Response: The text in the figure has been increased on Figure 3. It should be noted that the figures in the text are compressed jpeg files. The actual figures consist of high quality tif files, which can be zoomed in to see fine detail.

Comment: Figure 4: Similarly, an image of the shark showing the location of the sections would be helpful

Response: The approximate locations of the sections are now indicated on the CT models with arrowheads. We believe this is more useful for understanding the location of the slices than using an image of a whole shark.

Comment: Figure 6. The nano-CT images here are absolutely beautiful but too small to see clearly. They also need to be referenced in the figure caption and the color scheme given. 

Response: The CT images are referenced in the figure caption and the colour scheme is given below each of the models. The size of the colour scheme has been enlarged for clarity. The poor quality of the images in the word document are a result of compression the full detail can be seen in the TIF files which will be used as the figures for the text.

Comment: Figure 8: This is a nice, large image but the quality is not as good as the earlier stages. Why was this stage chosen for the full size image?

Response: This stage was displayed in this manner as there is no accompanying histological data. The images were from earlier renderings and have been replaced with models of the same quality as the rest of the embryos.

Comment: Figure 9: the images of fins are too small to see clearly. if this is the final figure size please reorganize the figure so that the images can be larger.

Response: The images of the fins have been enlarged for clarity.

Lines 93-94: missing words?

Response: The missing words have been reinserted.

Line 640: is "from" missing between shift and the?

Response: This has been corrected accordingly.

Lines 652-656: This sentence is hard to read. Please shorten/reword

Response: The sentence has been edited for clarity.

Line 662: in elephant sharks or a broader group?

Response: This point has been clarified see line 665.